# Longitudinal Measurements of Blood Biomarkers in Patients with Crohn’s Disease or Ulcerative Colitis Treated with Infliximab: Only the Latest Values in the Induction Period Predict Treatment Failure

**DOI:** 10.3390/jcm13040926

**Published:** 2024-02-06

**Authors:** Kim Oren Gradel, Bente Mertz Nørgård, Sonia Friedman, Jens Kjeldsen, Michael Due Larsen

**Affiliations:** 1Center for Clinical Epidemiology, Odense University Hospital, 5000 Odense, Denmark; bente.noergaard@rsyd.dk (B.M.N.); sfriedman1@tuftsmedicalcenter.org (S.F.); michael.d.larsen@ntnu.no (M.D.L.); 2Research Unit of Clinical Epidemiology, Department of Clinical Research, University of Southern Denmark, 5230 Odense, Denmark; 3Gastroenterology Division, Tufts Medical School, Tufts Medical Center, Boston, MA 02111, USA; 4Department of Medical Gastroenterology S, Odense University Hospital, 5000 Odense, Denmark; jens.kjeldsen@rsyd.dk; 5Research Unit of Medical Gastroenterology, Department of Clinical Research, University of Southern Denmark, 5230 Odense, Denmark; 6Department of Clinical and Molecular Medicine, Norwegian University of Science and Technology, 7034 Trondheim, Norway

**Keywords:** blood biomarkers, longitudinal measurements, infliximab, prediction, Crohn’s disease, ulcerative colitis

## Abstract

**Background:** Few studies have incorporated longitudinal assessments or used combinations of blood biomarkers as predictors of loss of response to biologic therapy for patients with Crohn’s disease (CD) or ulcerative colitis (UC). **Methods:** This is a population-based cohort study comprising Danish patients with CD or UC from 2008 to 2018. We used logistic regression to analyze whether levels and changes in levels of C-reactive protein (CRP), serum albumin, and hemoglobin, routinely measured during a 14-week infliximab induction period, predicted a change to another biologic medication or cessation of biologic therapy. **Results:** During the induction period, 2883 (1626 CD, 1257 UC) patients had 12,730, 12,040, and 13,538 specimens with CRP, serum albumin, and hemoglobin, respectively. In all, 284 patients (9.9%) switched to another biologic medication, and 139 (4.8%) ceased biologic therapy in the follow-up period. Only the most recent CRP and hemoglobin levels predicted the efficacy of infliximab treatment at approximately 14 weeks, a time point when the clinician often determines whether to continue treatment. **Conclusion:** Measurement of blood biomarkers prior to the clinical assessment does not predict the effectiveness of infliximab.

## 1. Introduction

Prediction of response to medical treatment is one of the main challenges in caring for patients with Crohn’s disease (CD) and ulcerative colitis (UC). This is particularly important for biologic medications because of their high cost and potentially serious side effects. Current guidelines recommend an individualized empiric strategy for handling the loss of response to biological treatment. This includes intensifying biologic therapy, changing within the class of biologic therapy, changing to another class of biologic therapy, optimizing concomitant treatment with conventional immunosuppressives, and finally, surgery [1,2]. This strategy assumes that it is preferable to completely exhaust one treatment option before discontinuing or changing the biologic therapy. Although tumor necrosis factor (TNF)-blocking therapy has resulted in long-term remission for CD and UC, up to 35% of the patients may have primary failure to this therapy [3]. Induction therapy for infliximab consists of three infusions: at day 0 and after 2 and 6 weeks, followed by maintenance therapy every 8 weeks. The effectiveness of the therapy is typically evaluated at the fourth treatment—that is, around 14 weeks after its initiation. If this induction therapy is not beneficial, whatever the reason, the clinician will change to an alternative therapy. An early prediction of treatment failure will facilitate the supervision of vulnerable patients and the decisions to change or supplement treatments earlier in the disease course. Whether an early prediction is possible can be assessed in longitudinal biomarker studies, preferably by combinations of biomarkers.

The blood biomarkers C-reactive protein (CRP), serum albumin, and hemoglobin are often used as standard biomarkers when starting and continuing biologic therapy. Among these, the inflammatory marker CRP has gained the most attention [3]. Serum albumin is a negative inflammatory biomarker [4] and a strong prognostic predictor in many diseases [5,6]. Anemia has a high prevalence in patients with inflammatory bowel disease (IBD) [7]. The hemoglobin level is included in prognostic indices for acute severe UC [8] and CD [9], and it has been evaluated as a prognostic predictor in a few studies [10].

There is no ideal blood biomarker, and the combination of two or more biomarkers has been recommended for prognostic studies [11]. Moreover, the vast majority of prognostic studies have only assessed one-time values of the biomarker, with a few exceptions for studies of CRP [12,13,14]. Studies with changes in CRP levels, rather than single measurements, have been recommended for patients with IBD [14], and the same may apply to serum albumin and hemoglobin.

CRP, serum albumin, and hemoglobin are routinely measured in patients with IBD. The little knowledge on whether their values, separately or in combination, could be used longitudinally for the prognosis thus prompted us to conduct this study.

In this population-based study, we examined whether levels of and changes in CRP, serum albumin, and/or hemoglobin during the 14-week induction period with infliximab treatment could predict a treatment failure, defined as a shift to another biologic therapy or cessation of biological treatment altogether.

## 2. Materials and Methods

### 2.1. Setting

This unselected study cohort is based on data from national Danish health registries. In Denmark (population approximately 5.8 million), all citizens have access to free public healthcare, and this enables us to develop a population-based study design based on an unselected nationwide study population [15]. The Danish healthcare system is tax-financed and thus free for the individual patient [16]. All patients with IBD are diagnosed and treated in public hospitals, with high completeness and validity of recorded IBD diagnoses and procedure codes [17,18]. Data from administrative registries can be linked by the unique personal identifier given to all Danish residents [15]. 

### 2.2. Study Population

We refer to Figure 1 for further details. 

In brief, the study population initially comprised patients in the Danish National Patient Registry [18] from which we retrieved patients treated with infliximab. In Denmark, anti-TNFs, interleukin inhibitors, and anti-integrin drugs are administered only in public hospitals or hospital-based out-patient settings, and procedure codes for each treatment and dates of administration are recorded in the Danish National Patient Registry [18]. The system holds a complete record of individuals receiving biological therapy and its associated consequences.

We restricted the inclusion of patients to those who fulfilled induction therapy (≥4 infliximab treatments) and ≥32 weeks of follow-up after the induction therapy period. We retrieved data for CRP, serum albumin, and hemoglobin in blood specimens from a laboratory database hosted by the Danish Health Data Authority, which covers all of Denmark as of 2008, except the Central Denmark Region (21% of Denmark’s population) [19]. The final study population included patients treated with infliximab, initiated between 7 January 2008 and 30 June 2018, and with blood specimen levels for CRP, serum albumin, or hemoglobin. From the Danish Civil Registration System [15], we retrieved data on the vital status up to 21 September 2020, including the date of death or emigration, if relevant.

### 2.3. Outcome

The outcome of this study was treatment failure, which we defined as either a shift to a biologic other than infliximab or cessation of biologic treatment altogether in the follow-up period. The follow-up spanned from 98 days (14 weeks) through 224 days (32 weeks) after the first-time infliximab treatment.

### 2.4. Statistical Analyses

We computed a contingency table with the patients’ baseline characteristics for all patients and stratified them into patients with CD or UC. Figure 2 gives the timeline for the study.

We included blood specimens for CRP, serum albumin, and hemoglobin, from 7 days (1 week) before through 98 days (14 weeks) after the first-time infliximab treatment. Because CRP was not normally distributed, we used its base-10 logarithm (CRP10), whereas serum albumin and hemoglobin values were not changed due to their normal distribution. Within this −7 to 98-day induction period, we focused on the earliest and the latest specimens and computed the days and changes in levels between these.

For each of the three biomarkers, we graphically depicted trajectories of daily mean levels in the −7 to 98-day period, separately for patients with and without treatment failure.

For the outcome, we computed logistic regression analyses with odds ratios (ORs) and 95% confidence intervals (CIs). These analyses were undertaken separately for CRP10, serum albumin, and hemoglobin and within each of these for the earliest and the latest levels, as well as for changes between these, divided them into percentiles (0–25%, 25–75%, and 75–100%). As the earliest levels contributed little to the outcome, these were omitted from the multivariate analyses in which we applied the following models:Model A: the latest level of CRP10, serum albumin, and hemoglobinModel B: Model A + changes in CRP10, serum albumin, and hemoglobin levelsModel C: Model A + changes in hemoglobin levelsModel D: the latest levels of CRP10 and hemoglobin + changes in hemoglobin levelsModel E: Model D + gender, age groups (0–16, 17–39, 40–59, ≥60 years), body mass index (BMI) (<18.5, ≥18.5 and <25, ≥25 and <30, ≥30 and <35, ≥35, missing), and quartiles of time from diagnosis of IBD to first-time treatment with infliximab (0–187, 188–808, 809–3053, 3054–14,652 days).

We computed areas under the receiver operating characteristic curves (AUROCs) for all the logistic regression analyses. For Models A–E, we compared these mutually by C-statistics [20].

Because of the retrospective nature of our data, in which values were not missing at random, we could not perform genuine longitudinal analyses [21]. Consequently, we reiterated all analyses in the following subgroups to assess the robustness of the data: (i) patients with CD; (ii) patients with UC; (iii) patients with 2 or more biomarker specimens in the −7 to 98 day period (the earliest and latest specimen comprised CRP10, serum albumin, and hemoglobin, and there were 1–83 days between the earliest and the latest specimen); (iv) as iii, but with 84–105 days between the earliest and latest specimen; (v) for patients with 3 or more biomarker specimens in the −7/98 day induction period, we replaced the latest level in the induction period by the level from the induction period’s 3rd biomarker specimen; (vi) where a change to another biologic drug was the only outcome, i.e., we skipped patients for whom biologic therapy was stopped altogether; (vii) for patients with UC, we divided the follow-up period into analyses before and from 1 April 2012 (date of approval of adalimumab); (viii) for all patients, we divided the follow-up period into analyses before and from 1 May 2014 (date of approval of vedolizumab).

In all analyses, a two-sided *p*-value of <0.05 was considered significant. The program Stata^®^, vs. 17, (StataCorp., College Station, TX, USA) was used for all analyses.

## 3. Results

A total of 2883 patients were included in the final study cohort (Figure 1), of whom 1626 (56.4%) had CD and 1257 (43.6%) had UC (Table 1).

There were no material differences between patients with CD and UC for any baseline characteristics in Table 1. Most of the patients had CRP, serum albumin, and hemoglobin measured, ranging from 92.8% for serum albumin for patients with UC to 99.9% for hemoglobin for the whole study population. In the period from 7 days before through 98 days after starting the infliximab treatment, each patient had a median (IQR) of 3 (2–5) specimens for CRP and serum albumin and 4 (3–5) for hemoglobin.

### 3.1. Trajectories of Mean Daily Levels 

During the day −7 to day 98 induction period, daily mean levels of CRP10 were generally higher and levels of serum albumin and hemoglobin were generally lower among patients with treatment failure in comparison to patients without treatment failure (Figure 3). 

The fitted lines deviated more from each other with the progression from day −7 to-ward day 98, most notably for hemoglobin, less for serum albumin, and least for CRP10. 

### 3.2. Logistic Regression Analyses Separately for the Three Biomarkers

In the six models, in which either the earliest or the latest biomarker level was the only covariate, only models with the latest level were significant (Table 2).

Hence, only the latest levels were combined with changes in the equivalent biomarker levels. In these models, the ORs (95% CIs) for the latest levels changed immaterially in comparison to the models without the changes. Changes were non-significant for CRP10 or serum albumin, although they were close to significant for serum albumin. For hemoglobin, changes were significant, with a trend of lower ORs with higher percentile changes. AUROCs ranged from 0.523 (for the earliest CRP10 level) to 0.594 (for the model with the latest hemoglobin level and hemoglobin changes). The results differed insignificantly between patients with CD and UC. 

### 3.3. Multivariate Logistic Regression Analyses

When the latest levels of each biomarker were combined (Model A), CRP10 and hemoglobin were significantly associated with the outcome (Table 3). 

With the amendment of changes in levels (Model B), only changes for hemoglobin were significant. We, therefore, excluded changes in levels for CRP10 and serum albumin (Model C) and further excluded the latest serum albumin level in Model D. ORs or 95% CIs did not change materially for the same covariates when these were compared between Models A, B, C, and D. The AUROC was 0.595 for Model A and very similar in Models B–D (ranging from 0.610 to 0.616). In the final Model E, the amendment of other possible confounders (gender, age, BMI, and time from diagnosis to first-time treatment with infliximab) did not change ORs or 95% CIs for the biomarker covariates in comparison to Model D (Table 3). Moreover, none of the amended confounders were significant.

### 3.4. AUROCs for Models A–E

In pairwise comparisons of AUROCs between the models, only Model A differed from Model E (*p* = 0.03), whereas the other comparisons were non-significant. 

### 3.5. Subgroup Analyses 

A total of 1800 patients had 2 or more biomarker specimens where both the earliest and latest specimens comprised CRP, serum albumin, and hemoglobin. The median time between the earliest and the latest specimen was 84 days, which we chose as a cut-off for dividing into patients with 1–83 (n = 867) and 84–105 days (n = 933). The results did not differ materially between these two groups or in comparison to the whole study population. 

A number of 1565 patients had 3 or more biomarker specimens, and 1440 of these (92.0%) could be used for the logistic regression analyses due to non-missing data for all the models’ variables. The results were very similar to those for the whole study population.

In the analyses where the outcome was restricted to change to another biologic drug than infliximab, all results were essentially the same as for the results in which stopping biologic treatment was included in the outcome definition.

The results of the analyses in the divided follow-up periods (before vs. after 1 April 2012 for patients with UC and before vs. after 1 May 2014 for all patients) did not deviate materially from the overall results either.

## 4. Discussion

We hypothesized that levels of and changes in the routinely retrieved biomarkers CRP, serum albumin, and hemoglobin could predict whether treatment with infliximab would be clinically valuable after its 14-week induction period. Ultimately, we hoped that the earliest measured levels could predict treatment failure sooner. However, statistical significance was seen only for the latest measured CRP and hemoglobin levels in the induction period and for changes in the hemoglobin levels. These latest measurements occurred at a time point when the clinician will likely assess the patient’s condition anyway, including decisions on whether treatment with infliximab needs to continue or change to another drug, which we defined as treatment failure. Moreover, AUROCs below 0.7, regardless of the model, reflected non-acceptable discrimination [22].

Approximately one-third of patients will be non-responders to infliximab (and other biologics) and will require alternative treatment. It would be ideal if biomarkers collected at the beginning of the biologic therapy period could predict a clinical response. Patients would be able to avoid a prolonged flare and further clinical decline by choosing a medication other than infliximab. This large study finds, rather, that patients must undergo a full infliximab induction, and only biomarker measurements around 14 weeks will be reflective of clinical response. CRP and hemoglobin are helpful insofar as they are reflective of the clinical response to infliximab induction and can help differentiate common concurrent diagnoses such as irritable bowel syndrome or chronic pain syndrome, which may obscure clinical response.

The fecal calprotectin level is the gold standard for assessing the severity of IBD [3,14]. Our real-life data were too sparse to incorporate this specimen type in the prognostic assessment, as only 993 patients (34.4%) had fecal calprotectin specimens, and among these, 578 (58.2%) had one specimen only, which hampered longitudinal analyses. Consequently, the study was based on CRP, serum albumin, and hemoglobin from blood specimens, which are normally retrieved concomitantly with the infliximab treatment. Still, half of the patients received their fourth treatment after the 14-week induction period, some up to 32 weeks thereafter. This was also reflected in varying numbers of biomarker specimens in the day −7 to day 98 period, to which we restricted our analyses to minimize heterogeneity.

We were inspired by a previous study, in which CRP levels ≥10 mg/L or a clinical score ≥5 (the Harvey–Bradshaw index for CD and the Simple Clinical Colitis Activity Index for UC) predicted a nearly fourfold increased risk of steroid therapy or surgery after the 14-week induction therapy period [23]. In the present study, we focused on the utility of longitudinal aspects of CRP, serum albumin, and hemoglobin, as well as of their combinations. This was also the reason for not incorporating clinical scores, which were recorded for the end of the induction therapy period and, therefore, did not contribute to an earlier prognostic assessment.

Most biomarker studies in patients with IBD are either cross-sectional, in which biomarkers were compared with clinical scores [24], or one-time levels that predicted future adverse outcomes, such as complications after surgery [25], increased risk of surgery or medication [26], prolonged hospitalizations [27], or steroid non-response [28]. Newer studies have incorporated one-time levels of CRP and serum albumin, often as the CRP/serum albumin ratio (CAR) [24]. To assess the separate impact of CRP, serum albumin, and hemoglobin, we did not use CAR in our study, and because serum albumin had little impact in models with hemoglobin, it was skipped in the final analyses. Fewer studies have evaluated changes in biomarker levels as predictors of an adverse outcome. Studies from European countries have incorporated CRP levels at baseline, after 4/10/14 weeks, and the difference or ratio between these, for patients with UC [29] or CD [12,13,30,31,32], all treated with infliximab. Comparison to our results is difficult due to the various follow-up periods and outcomes or the rates only computed for patients with high first-time CRP levels. A Belgian study of 614 patients with CD found that the baseline CRP level did not predict a sustained clinical benefit of infliximab during a 5-year follow-up period, whereas a drop of >50% from the baseline level or normalization to <3 mg/L did [32]. This is in accordance with our results and supports the recommendation of longitudinal assessments of biomarker levels [14]; however, no AUROCs were reported in the Belgian study. A Japanese study of 72 UC patients used the ratio between the week 0 and 2 levels of both CRP and serum albumin to predict the response to infliximab at week 14 [33]. The ratio of CRP, but not of serum albumin, was a predictor with a good discriminatory ability (AUROC = 0.799). A few older studies from the UK have reported longitudinal data for serum albumin and hemoglobin, but the low number of patients (50 or fewer) precludes firm conclusions [34,35,36,37,38]. In our study, there were no differences between patients with UC and CD. For CRP, this is not in accordance with several studies, which state that a high CRP response is mainly seen in patients with CD but not with UC [3]. For hemoglobin, the reverse has been reported, i.e., it is prognostic in patients with UC but not with CD [10]. Comparison between our results and other studies is difficult due to different settings and outcomes and the few studies that have assessed longitudinal aspects. 

Our study is population-based and includes a high number of patients, both with UC and CD. Among the studies with longitudinal analyses, the highest number of patients was 718 [31], and we have not encountered other studies that combined CRP, serum albumin, and hemoglobin. Both the diagnoses of IBD and the procedure recordings of infliximab treatment have high completeness and validity [17,18]. The outcome is relevantly based on the physicians’ assessments after the 14-week induction therapy period.

Our study also has limitations that deserve further consideration. First, real-life data with varying numbers of specimens and time intervals between these hamper genuine longitudinal analyses [21]. However, stratifying the analyses into 1–83 and 84–105 days between the earliest and latest specimens did not change our results materially. Moreover, ORs changed little between various models or when the last specimen was replaced by the third. Thus, our results seemed robust despite the heterogeneous data. Second, we do not know to what degree the outcome (i.e., the physicians’ decisions) was based on the biomarker results. Hence, possible predictors may be part of the outcome, although the magnitude of this is difficult to quantify for a holistic clinical assessment. However, this pitfall would have been a bigger problem if the results had shown that the biomarkers were strong predictors. Third, the inclusion of fecal calprotectin in the analyses would be beneficial, but the number of specimens did not allow this. Fourth, as in other retrospectively derived data, there are unknown confounders such as intake of other drugs, smoking, or alcohol intake. The intake of other drugs may, however, be related to exacerbations of IBD symptoms, which are also correlated to levels of CRP, serum albumin, and hemoglobin, so confounding by indication is a pitfall if other drugs than infliximab are included in the analyses. Moreover, the amendment of possible confounders in Model E did not change the ORs for the biomarkers or the AUROCs materially, and it is unlikely that this would differ for other possible confounders. Finally, 18 patients underwent gastrointestinal surgery in the induction treatment period, which is an undesirable outcome regardless of the decision regarding infliximab continuation. These patients had more biomarker specimens (median 19.5, IQR 7–32), and 16 (88.9%) experienced the outcome, but their exclusion did not change the analyses materially.

## 5. Conclusions

Results from specimens of CRP, serum albumin, and hemoglobin retrieved before the end of the 14-week induction treatment period with infliximab were weak predictors of whether infliximab treatment should be continued thereafter. Additional studies, including genetics, serology, and correlation with specific IBD subtypes, need to be performed in order to find more clinically predictive biomarkers.

## Figures and Tables

**Figure 1 jcm-13-00926-f001:**
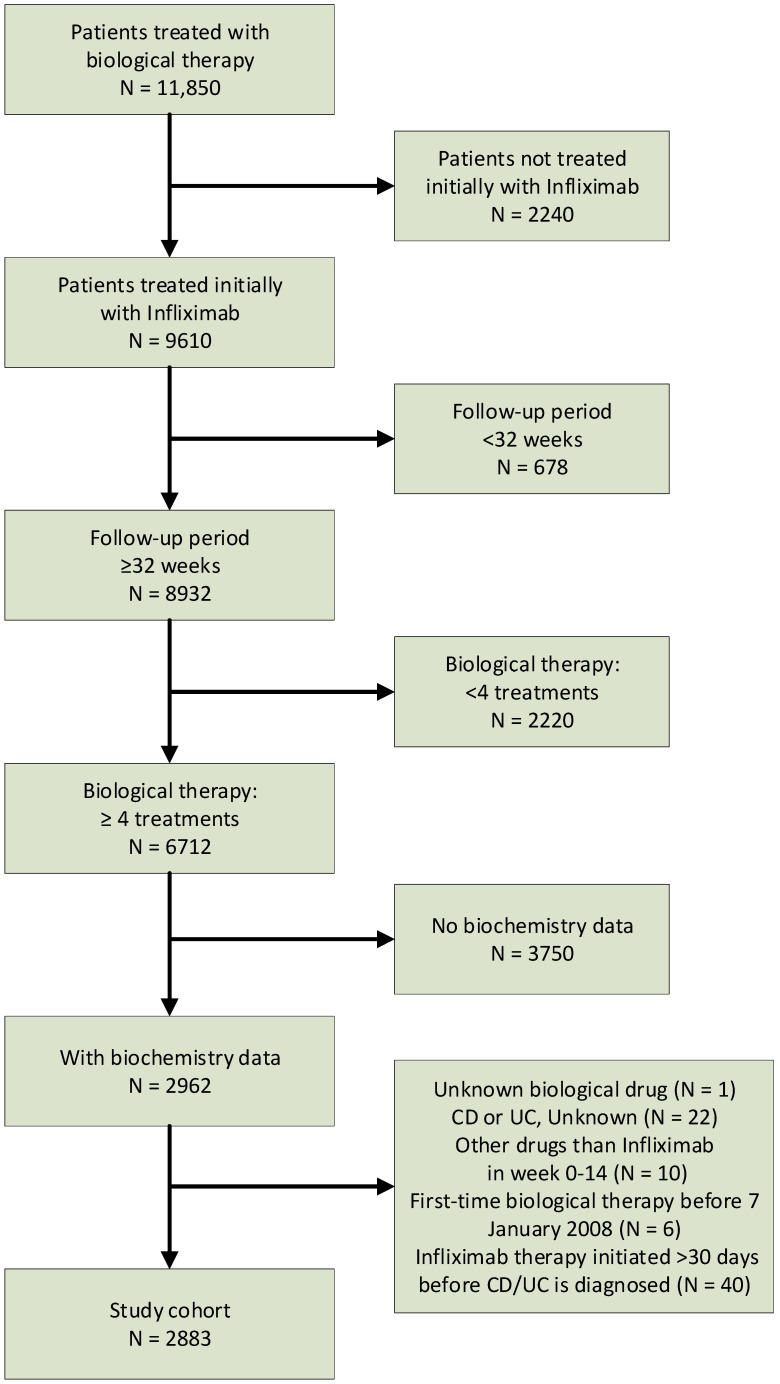
Derivation of the study population.

**Figure 2 jcm-13-00926-f002:**
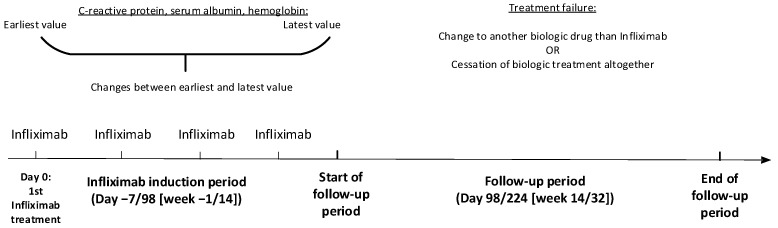
Timeline for the study.

**Figure 3 jcm-13-00926-f003:**
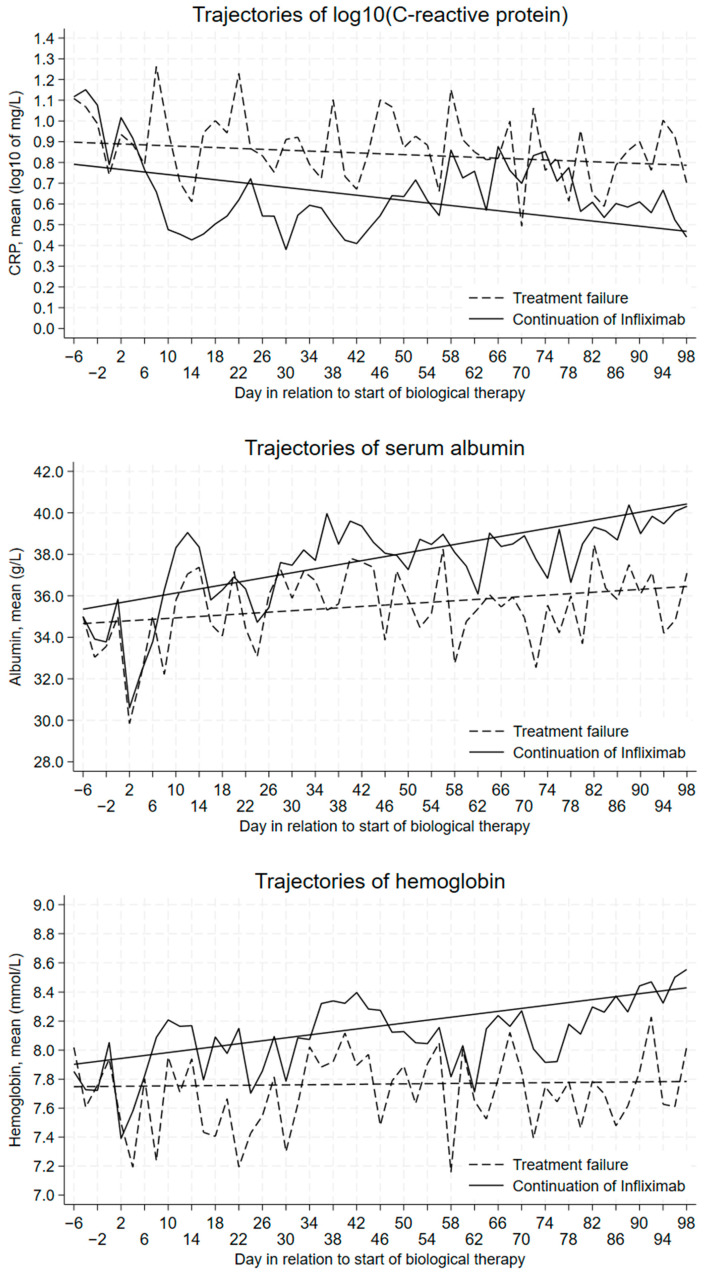
Daily mean levels of biomarkers, with fitted lines, days −7/98 in relation to starting biological therapy.

**Table 1 jcm-13-00926-t001:** Descriptive characteristics of patients with Crohn’s disease or ulcerative colitis introduced to infliximab therapy from 2008 through 2018.

Characteristic	All Patients (n = 2883)	Crohn’s Disease (n = 1626)	Ulcerative Colitis (n = 1257)
Gender			
Females	1487 (51.6)	853 (52.5)	634 (50.4)
Males	1396 (48.4)	773 (47.5)	623 (49.6)
Age, years ^1^			
Range	1.1–89.3	1.1–82.3	3.7–89.3
Median (IQR)	34.0 (23.0–47.6)	30.9 (21.6–46.2)	36.9 (25.5–50.0)
Body mass index ^1^			
<18.5	52 (1.8)	38 (2.3)	14 (1.1)
≥18.5, <25	544 (18.9)	310 (19.1)	234 (18.6)
≥25, <30	329 (11.4)	177 (10.9)	152 (12.1)
≥30, <35	142 (4.9)	84 (5.2)	58 (4.6)
≥35	62 (2.2)	30 (1.9)	32 (2.6)
Missing	1754 (60.8)	987 (60.7)	767 (61.0)
C-reactive protein, measured ^2^			
Patients	2817 (97.7)	1594 (98.0)	1223 (97.3)
Number of values			
All	12,730	6208	6522
Per patient			
Range	1–51	1–51	1–45
Median (IQR)	3 (2–5)	3 (2–4)	4 (3–6)
Serum albumin, measured ^2^			
Patients	2709 (94.0)	1543 (94.9)	1166 (92.8)
Number of values			
All	12,040	5876	6164
Per patient			
Range	1–52	1–52	1–37
Median (IQR)	3 (2–5)	3 (2–4)	4 (3–6)
Hemoglobin, measured ^2^			
Patients	2880 (99.9)	1624 (99.9)	1256 (99.9)
Number of values			
All	13,538	6594	6944
Per patient			
Range	1–49	1–49	1–46
Median (IQR)	4 (3–5)	2 (3–5)	4 (3–7)
Gastrointestinal surgery ^3^	18 (0.6)	8 (0.5)	10 (0.8)
Shift to another biological drug ^4^	284 (9.9)	140 (8.6)	144 (11.5)
Ceasing biological treatment ^4^	139 (4.8)	56 (3.4)	83 (6.6)

^1^ On the date of initiating biological therapy. ^2^ From 7 days (1 week) before through 98 days (14 weeks) after initiating biological therapy. ^3^ Nordic Classification of Surgical Procedures, codes KJFH* (total colectomy) or KJFB* (intestinal resection), from 7 days (1 week) before through 98 days (14 weeks) after initiating biological therapy. ^4^ From 98 days (14 weeks) through 224 days (32 weeks) after initiating biological therapy.

**Table 2 jcm-13-00926-t002:** Odds ratio (OR) and area under the receiver operating characteristic curve (AUROC) for the biomarker levels and their changes as predictors for treatment failure.

Model	OR (95% CI)	AUROC ^1^
log10 of CRP ^1^, earliest	1.12 (0.94–1.34)	0.523
log10 of CRP, latest	1.72 (1.39–2.13)	0.568
Serum albumin, earliest	0.98 (0.96–1.00)	0.529
Serum albumin, latest	0.95 (0.93–0.97)	0.570
Hemoglobin, earliest	0.90 (0.81–1.00)	0.528
Hemoglobin, latest	0.74 (0.67–0.83)	0.575
log10 of CRP, 0–25 percentile change	1 (ref.)	0.572
log10 of CRP, 25–75 percentile change	0.92 (0.68–1.26)	
log10 of CRP, 75–100 percentile change	0.90 (0.68–1.21)	
log10 of CRP, latest	1.76 (1.41–2.20)	
Serum albumin, 0–25 percentile change	1 (ref.)	0.576
Serum albumin, 25–75 percentile change	0.76 (0.57–1.02)	
Serum albumin, 75–100 percentile change	0.73 (0.52–1.02)	
Serum albumin, latest	0.95 (0.93–0.98)	
Hemoglobin, 0–25 percentile change	1 (ref.)	0.594
Hemoglobin, 25–75 percentile change	0.74 (0.57–0.96)	
Hemoglobin, 75–100 percentile change	0.60 (0.44–0.82)	
Hemoglobin, latest	0.78 (0.70–0.88)	

^1^ The base-10 logarithm of C-reactive protein.

**Table 3 jcm-13-00926-t003:** Logistic regression analyses, combined for the three biomarkers, in addition to adjustment for gender, age, body mass index, and time between diagnosis and start of infliximab treatment.

Cofactor	Model A (0.595) ^1^	Model B (0.616)	Model C (0.614)	Model D (0.610)	Model E (0.623)
log10 of CRP, latest	1.43 (1.12–1.82)	1.41 (1.09–1.82)	1.39 (1.09–1.77)	1.48 (1.19–1.85)	1.49 (1.18–1.87)
Serum albumin, latest	0.98 (0.95–1.01)	0.98 (0.95–1.01)	0.98 (0.95–1.01)		
Hemoglobin, latest	0.80 (0.70–0.91)	0.85 (0.74–0.98)	0.84 (0.73–0.96)	0.83 (0.73–0.94)	0.81 (0.71–0.93)
log10 of CRP, 0–25 percentile difference		1 (reference)			
log10 of CRP, 25–75 percentile difference		0.95 (0.67–1.35)			
log10 of 75–100 percentile difference		0.93 (0.65–1.32)			
Serum albumin, 0–25 percentile difference		1 (reference)			
Serum albumin, 25–75 percentile difference		0.85 (0.62–1.19)			
Serum albumin, 75–100 percentile difference		0.98 (0.65–1.47)			
Hemoglobin, 0–25 percentile difference		1 (reference)	1 (reference)	1 (reference)	1 (reference)
Hemoglobin, 25–75 percentile difference		0.74 (0.55–0.99)	0.73 (0.55–0.97)	0.71 (0.54–0.93)	0.70 (0.53–0.93)
Hemoglobin, 75–100 percentile difference		0.54 (0.37–0.80)	0.56 (0.39–0.80)	0.56 (0.40–0.78)	0.57 (0.41–0.80)
Males					0.96 (0.75–1.23)
Age, 0–16 years					1 (reference)
Age, 17–39 years					1.00 (0.57–1.76)
Age, 40–59 years					1.48 (0.82–2.68)
Age, ≥60 years					1.33 (0.69–2.55)
Body mass index, <18.5					0.55 (0.19–1.62)
Body mass index, ≥18.5 and <25					1 (reference)
Body mass index, ≥25 and <30					0.82 (0.54–1.25)
Body mass index, ≥30 and <35					0.98 (0.57–1.67)
Body mass index, ≥35					0.59 (0.25–1.38)
Body mass index, missing					0.78 (0.58–1.05)
Time, diag-biol ^2^, 0–187 days					1 (reference)
Time, diag-biol, 188–808 days					1.15 (0.83–1.59)
Time, diag-biol, 809–3053 days					1.07 (0.76–1.50)
Time, diag-biol, 3054–14,652 days					1.00 (0.70–1.42)

^1^ Brackets: area under the receiver operating characteristic curve for the model, based on the 2103 patients in Models A–C (in order to enable comparisons between the models). ^2^ Time from diagnosis of inflammatory bowel disease to start of infliximab treatment.

## Data Availability

This register-based non-interventional study follows the STROBE guideline [39]. The authors of this paper have no special access privileges to the data used in this study, and other researchers may apply for access to the data through an application to the Research Service at the Danish Health Data Authority.

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
