# Peer review of "Longitudinal Measurements of Blood Biomarkers in Patients with Crohn’s Disease or Ulcerative Colitis Treated with Infliximab: Only the Latest Values in the Induction Period Predict Treatment Failure"

_jcm, 2024, doi:10.3390/jcm13040926_

Round 1

Reviewer 1 Report

Comments and Suggestions for Authors

·         The original article of “Longitudinal measurements of blood biomarkers in patients with Crohn's Disease or Ulcerative Colitis treated with infliximab”: of definitely of interest, but more work is needed to do the research field and included data justice.

·         There are many typographical errors in the text. English editing by a native English speaker is recommended.

·         The type of analysis method should also be mentioned in the abstract.

·         The keywords used in the abstract are very poorly chosen.

·         The introduction is very poorly written and there is no mention of how and why the study was conducted. No important information about the relationship between blood biomarker and predict a treatment failure can be deduced from the introduction, and there is no literature on the effect of the blood biomarker in the introduction.

·         What will be the compliance rate of the reviewed drug to complete the study?

·         Many factors can affect the results as possible confounders, but the author did not pay attention to them. For example, intake of other drug such as MTX or AZARAM  received, family history of disease, duration of disease, smoking, alcohol intake, weight change, menopausal status,….. These items should be adjusted in the analysis.

·         How is the compliance of people with the drug in question measured?

·         I don't understand why the authors focused on these biomarkers. While calprotectin and platelet levels are among the most important biomarkers of treatment response. These things should be taken into account

·         On what basis is the drug dose and the duration of the intervention determined?

·         How was the sample size calculated?

·         Why are patients not examined in terms of clinical symptoms and disease activity based on disease-related indicators?

·         ​In order to examine the response or non-response to treatment with biological drugs, the authors should examine drug levels as well as antibody levels of biological drugs, and this is a major limitation of this study?

·         Were the participants not taking any medication before or during the study?

·         Why is there no dietary and physical activity assessment?

·         The discussion should fully discuss the clinical effectiveness and significance of the results based on clinical criteria.

Comments on the Quality of English Language

·       

·         There are many typographical errors in the text. English editing by a native English speaker is recommended.

·        

Author Response

Reviewer 1:

The original article of “Longitudinal measurements of blood biomarkers in patients with Crohn's Disease or Ulcerative Colitis treated with infliximab”: of definitely of interest, but more work is needed to do the research field and included data justice.

There are many typographical errors in the text. English editing by a native English speaker is recommended.

Answer: Thank you for this. One of the co-authors (Sonia Friedman) is a native English speaker and the typographical errors have been corrected in the revised manuscript.

The type of analysis method should also be mentioned in the abstract.

Answer: Thank you for this. We have now mentioned in the abstract that we used logistic regression analyses.

The keywords used in the abstract are very poorly chosen.

Answer: Thank you for this. We have deleted “treatment failure” and added the keywords “Crohn’s Disease” and “ulcerative colitis”.

The introduction is very poorly written and there is no mention of how and why the study was conducted.

Answer: We do not believe an introduction should encompass how a study was conducted, as this is described in the methods section. However, we agree that it is important to emphasize why it was conducted. We have amended the following section to the introduction: “CRP, serum albumin, and hemoglobin are routinely measured in patients with IBD. The little knowledge on whether their values, separately or in combination, could be used longitudinally for the prognosis, thus prompted us to conduct this study.”     

No important information about the relationship between blood biomarker and predict a treatment failure can be deduced from the introduction, and there is no literature on the effect of the blood biomarker in the introduction.

Answer: This is stated in the discussion section where the relevant literature of the field has been discussed. Its inclusion in the introduction section would be superfluous reiterations. If the editor disagrees, we will gladly transfer some of this to the introduction section. 

What will be the compliance rate of the reviewed drug to complete the study?

Answer: As infliximab was given in the hospital under the physician’s supervision the compliance rate is expected to be close to 100%. The drug cannot be administered without recording it in the hospital’s electronic medical system and these data are transferred to the Danish National Patient Registry from which we received the data.   

Many factors can affect the results as possible confounders, but the author did not pay attention to them. For example, intake of other drug such as MTX or AZARAM  received, family history of disease, duration of disease, smoking, alcohol intake, weight change, menopausal status,….. These items should be adjusted in the analysis.

Answer: We agree that there are unknown confounders as in all epidemiological studies using real-world registry data and we have now added this as a limitation to the study:

“Fourthly, as in other retrospectively derived data there are unknown confounders such as intake of other drugs, smoking, or alcohol intake. The intake of other drugs may, however, be related to exacerbations of IBD symptoms, which are also correlated to levels of CRP, serum albumin, and hemoglobin, so confounding by indication is a pitfall if other drugs than infliximab are included in the analyses. Moreover, the amendment of confounders in Model E changed the ORs for the biomarkers and the AUROCs immaterially and it is unlikely that this would differ for other confounders.”

The reviewer mentions disease duration as an example of a confounder, but we have adjusted for that (cf. Table 3, Model E).

 How is the compliance of people with the drug in question measured?

Answer: Please, see the answer to the above remark (“What will be the compliance rate of the reviewed drug to complete the study?”).

I don't understand why the authors focused on these biomarkers. While calprotectin and platelet levels are among the most important biomarkers of treatment response. These things should be taken into account.

Answer: We focused on data, which are routinely measured in the clinic. We have underlined this in the manuscript and further added it to the abstract:

We used logistic regression to analyze whether levels and changes in levels of C-reactive protein (CRP), serum albumin, and hemoglobin, routinely measured during a 14-week infliximab induction period, predicted a change to another biologic medication or cessation of biologic therapy.

As stated, few patients had calprotectin results, which did not enable longitudinal analyses. 

On what basis is the drug dose and the duration of the intervention determined?

Answer: Our study is non-interventional and all the patients have received standard care according to the current guidelines.

How was the sample size calculated?

Answer: We did not calculate the sample size, but used unselected nation-wide data encompassing all treatments with infliximab during the study period. As the study was population-based we included all patients who fulfilled the study criteria during the study period (7 January 2008-30 June 2018). We believe Figure 1 gives a good overview of how the study population was derived, but suggestions for improvements are of course welcome.

Why are patients not examined in terms of clinical symptoms and disease activity based on disease-related indicators?

Answer: Disease activity was not assessed routinely prior to the end of the induction period [1]. At the 14-week examination, the disease activity played an important role for the outcome, that is, the physician’s decision on whether to continue the treatment with infliximab. We refer to the following from the discussion section:

“This was also the reason for not incorporating clinical scores, which were recorded for the end of the induction therapy period and therefore did not contribute to an earlier prognostic assessment.”       

​In order to examine the response or non-response to treatment with biological drugs, the authors should examine drug levels as well as antibody levels of biological drugs, and this is a major limitation of this study?

Answer: This would require a new study with a different outcome. We believe the physician’s decision of whether infliximab was effective in the induction period is a valid measure, regardless of levels of infliximab or its antibodies.  

Were the participants not taking any medication before or during the study?

Answer: As regards infliximab, all patients were bio naive at the beginning of the induction period. Moreover, see our answer to the remark above on confounders (“Many factors can affect the results as possible confounders, but the author did not pay attention to them. For example, intake of other drug such as MTX or AZARAM  received, family history of disease, duration of disease, smoking, alcohol intake, weight change, menopausal status,….. These items should be adjusted in the analysis.”).

Why is there no dietary and physical activity assessment?

Answer: These data are not recorded in our registries, but all patients have received standard treatment, including advice on dietary and physical activity. The outcome is a holistic assessment of whether infliximab was effective or not in the induction period. The impact of specific factors on the physician’s decision is difficult to assess, which we have also mentioned as a limitation in the discussion section:

“Secondly, we do not know to what degree the outcome (i.e. the physicians’ decision) was based on the biomarker results. Hence, possible predictors may be part of the outcome, although the magnitude of this is difficult to quantify for a holistic clinical assessment. However, this pitfall would have been a bigger problem if the results had shown that the biomarkers were strong predictors.”.

The discussion should fully discuss the clinical effectiveness and significance of the results based on clinical criteria.

Answer: The study concludes that CRP, serum albumin, or hemoglobin levels are not effective in predicting the outcome prior to the end of the induction period. The clinical criteria are part of the outcome, i.e. the physician’s decision at the end of the induction period. We think this has been clearly described in the discussion:

“However, statistical significance was seen only for the latest measured CRP and hemoglobin levels and for changes in the latter. These latest measurements occurred at a time point where the clinician will assess the patient’s condition anyway, including decisions on whether treatment with infliximab needs to cease or change to another drug, which we defined as treatment failure. Moreover, AUROCs below 0.7, regardless of the model, reflected non-acceptable discrimination [22].”

If the editor disagrees, we will gladly elaborate further on this.   

Reference:

  1. Larsen MD, Nørgård BM, Kjeldsen J: Does Disease Activity After Induction Treatment With Biologics Predict Short-Term Outcome in Crohn's Disease and Ulcerative Colitis? Inflamm Bowel Dis 2022.

Reviewer 2 Report

Comments and Suggestions for Authors

The manuscript presents a carefully done study of a very complex problem, which was not solved, probably because of the lack of attention to the mechanisms of pathologies and therapies, which are not at all mentioned in the manuscript. Potential biomarkers for early diagnostics of therapeutic success were desired to be determined, but the study resulted in a negative outcome, i.e. the desired biomarkers were not established. This reviewer agrees with the authors’ formulation of this outcome in Discussion (lines 30-36).  If this negative outcome is chosen to be published, it deserves a more direct mentioning in the title, like: CRP or HB after the first treatment do not predict a success in…

Besides, the whole presentation must be much more clear and better structured to highlight the points mentioned in the discussion. Probably some essential parts must be separated from secondary analyses, the latter transferred to supplementary materials.

If the goal is the early diagnosis, the authors should focus on this diagnosis, instead of the longitudinal follow-up, which is suitable for something else. There is a contradiction between this goal and the current study design. Probably, therefore the goal is not clearly formulated. In this regard, I would suggest to try some ratios between the parameters of the biochemical blood analyses presumably available for the patients – for instance, compare ratios CRP/SA, SRP/Hb and all available combinations (CRP/Na+), CRP/transaminases etc. for each patient at the early time point. Sometimes such ratios have more discriminating power. Many inflammations are associated with decreased levels of NAD+; thiamine is known to help the states associated with the inflammation and bowel disease. So, NAD+ and thiamine could be applicable to follow the therapy. The focus of experimental design on the chosen indicators is not well justified, although there are of course limitations due to retrospective nature of the study.

Clear presentation and logic are especially important when you would like to publish negative outcome.

Specific comments are listed below.

referring to “the latest measurements” in the complicate longitudinal studies should be substituted for the clearly defined time.

“…biologic agents, because of their high cost and potentially serious side effects…”

Definition of “biologic agent” is required. Biologic ally produces substances, e.g., have less restrictions to be used as nutrients – hence, the statement is now quite exact.

Short information on the (presumed) mechanism of action of infliximab is required in the introduction, as from the beginning one only may deduce that infliximab belongs to “biologic agents”, which is not at all deducible from Figure 1.

Line 70: abbreviation IBS is deciphered only later in the method section. To make the perception of the text easier, it is better not to use this and other abbreviations of the pathologies (except in the figures where it may be deciphered in the legend). Аr instance, in Table 1 the authors do not use the abbreviations, and making this consistent would benefit follow-up of the text. By the way, this Table does not use CRP, SA and Hb either, that is welcome. Especially SA is less intuitive in this row.

Lines 134-136 – relationship/justification of the logarithmic transformation of the not normal distribution in CRP values should be introduced.

Lines 167-169 – unclear

Line 170 – needs editing

Lines 196-198 – this should be expected, shouldn’t it? 

Figure 2 has a font whose size is too small

Author Response

Reviewer 2:

The manuscript presents a carefully done study of a very complex problem, which was not solved, probably because of the lack of attention to the mechanisms of pathologies and therapies, which are not at all mentioned in the manuscript. Potential biomarkers for early diagnostics of therapeutic success were desired to be determined, but the study resulted in a negative outcome, i.e. the desired biomarkers were not established. This reviewer agrees with the authors’ formulation of this outcome in Discussion (lines 30-36).  If this negative outcome is chosen to be published, it deserves a more direct mentioning in the title, like: CRP or HB after the first treatment do not predict a success in…

Answer: We think that we highlight the main findings in the title: “only the latest values in the induction period predict treatment failure“.

Besides, the whole presentation must be much more clear and better structured to highlight the points mentioned in the discussion. Probably some essential parts must be separated from secondary analyses, the latter transferred to supplementary materials.

Answer: Thank you for this. The resubmitted version has been scrutinized by a native English speaker and we have clarified the text according to the reviewer’s specific suggestions (see below).

If the goal is the early diagnosis, the authors should focus on this diagnosis, instead of the longitudinal follow-up, which is suitable for something else. There is a contradiction between this goal and the current study design.

Answer: The goal was not an early diagnosis, but the prediction of the assessment after the 14-week induction period. If our results had been positive, this would have helped in the surveillance of patients during the induction period, but the final decision on whether infliximab was effective would still take place thereafter. We believe it was important to evaluate changes in the biomarker levels in the prediction models, which has also been done for other diseases [2].  

Probably, therefore the goal is not clearly formulated. In this regard, I would suggest to try some ratios between the parameters of the biochemical blood analyses presumably available for the patients – for instance, compare ratios CRP/SA, SRP/Hb and all available combinations (CRP/Na+), CRP/transaminases etc. for each patient at the early time point. Sometimes such ratios have more discriminating power. Many inflammations are associated with decreased levels of NAD+; thiamine is known to help the states associated with the inflammation and bowel disease. So, NAD+ and thiamine could be applicable to follow the therapy. The focus of experimental design on the chosen indicators is not well justified, although there are of course limitations due to retrospective nature of the study.

Answer: Thank you for these suggestions. In this registry-based study we have used the commonly measured biomarkers, so ratios between CRP, serum albumin, and hemoglobin are possible whereas we do not have data for the other suggested biomarkers. The ratio between CRP and serum albumin is applied in many studies, but as serum albumin was non-significant we did not see any reason to proceed with this.       

Clear presentation and logic are especially important when you would like to publish negative outcome.

Answer: We agree and hope that the clarifications in the resubmitted manuscript have improved the clarity.

Specific comments are listed below.

referring to “the latest measurements” in the complicate longitudinal studies should be substituted for the clearly defined time.

Answer: We refer to Figure 2, which shows that the latest measurement occurs within the 14-week induction period. A clearly defined time is difficult to report as this may vary. In the discussion section we have clarified this:

“However, statistical significance was seen only for the latest measured CRP and hemoglobin levels in the induction period and for changes in the latter.”

(where “in the induction period” has been amended). 

“…biologic agents, because of their high cost and potentially serious side effects…”

Definition of “biologic agent” is required. Biologic ally produces substances, e.g., have less restrictions to be used as nutrients – hence, the statement is now quite exact.

Answer: We agree with this and have used other terms (biologic therapy/treatment/drugs/medications) in the resubmitted manuscript.

Short information on the (presumed) mechanism of action of infliximab is required in the introduction, as from the beginning one only may deduce that infliximab belongs to “biologic agents”, which is not at all deducible from Figure 1.

Answer: We believe our target readers will be gastroenterologists for whom infliximab is familiar. Our reference 3 (Tarapatzi et al., J. Gastrointestin. Liver Dis. 2022, 31, 229-243, doi:10.15403/jgld-4229) is a nice review of mechanisms of biological drugs.

Line 70: abbreviation IBS is deciphered only later in the method section. To make the perception of the text easier, it is better not to use this and other abbreviations of the pathologies (except in the figures where it may be deciphered in the legend). Аr instance, in Table 1 the authors do not use the abbreviations, and making this consistent would benefit follow-up of the text. By the way, this Table does not use CRP, SA and Hb either, that is welcome. Especially SA is less intuitive in this row.

Answer: Thank you for this. We have explained what IBD stands for at its first occurrence. We have changed SA to serum albumin and Hb to hemoglobin throughout the article, but maintained CRP and CRP10 as these are too tedious to reiterate.   

Lines 134-136 – relationship/justification of the logarithmic transformation of the not normal distribution in CRP values should be introduced.

Answer: We have written the following:

“Because CRP was not normally distributed, we used its base-10 logarithm (CRP10)”

It is unclear to us why further justifications are needed.

Lines 167-169 – unclear

Answer: We presume the reviewer refers to the following:

“v) for patients with 3 or more biomarker specimens in the −7/98 day period, we replaced the latest level in the analyses with the levels from the 3rd biomarker specimen”

We have hopefully clarified it by the following:

“v) for patients with 3 or more biomarker specimens in the −7/98 day induction period, we replaced the latest level in the induction period by the level from the induction period’s 3rd biomarker specimen”

Line 170 – needs editing

Answer: We presume the reviewer refers to the following:

“vi) where a change to another biological drug was the outcome, i.e. we skipped patients for whom biological therapy was stopped altogether.”

We have hopefully clarified it by the following:

“vi) where a change to another biological drug was the only outcome, i.e. we skipped patients for whom biological therapy was stopped altogether.”

Lines 196-198 – this should be expected, shouldn’t it? 

Answer: Yes, but it was not significant and consequently could not predict our outcome. 

Figure 2 has a font whose size is too small

Answer: We have enlarged the font size in Figure 2.

Reference:

  1. Póvoa P, Garvik OS, Vinholt PJ, Pedersen C, Jensen TG, Kolmos HJ, Lassen AT, Gradel KO: C-reactive protein and albumin kinetics after antibiotic therapy in community-acquired bloodstream infection. Int J Infect Dis 2020, 95:50-58.

Reviewer 3 Report

Comments and Suggestions for Authors

This study on the predictive capacity of 3 routinely used and widely available analytical markers in clinical practice is relevant, novel and employs appropriate methods. However, it starts from a premise that may not be completely true. The authors collect data from patients with inflammatory bowel disease treated with infliximab in the period 2008 to 2018. The criterion for failure to respond to treatment with infliximab consisted of a shift to another biological drug than infliximab or ceasing biological treatment altogether in the follow-up period of 32 weeks. Any patients who continued on infliximab beyond 32 weeks along the 10-year study period were considered to have responded to treatment, which is not necessarily true.

Adalimumab, the second biologic drug approved to treat Crohn's disease after infliximab, was approved in Europe for use in Crohn's disease in April 2007, but for ulcerative colitis, adalimumab was not approved by the European Union in April 2012. The next drug (vedolizumab) was not approved by the European Medicines Agency for both Crohn's disease and ulcerative colitis until May 2014. Finally, ustekinumab was approved in Europe to treat Crohn's disease in May 2016, and for ulcerative colitis in September 2019. Before these dates, biological alternatives to infliximab were simply not available for use in Europe.

Therefore, it seems logical to think that some patients included in the registry supporting this study could have maintained treatment with infliximab beyond 32 weeks, even if the effectiveness had been suboptimal, simply because there was no therapeutic alternative available at the moment they were treated. Taking this limitation into account, it would be necessary for the authors to develop subgroup analyses, considering various time frames and types of inflammatory bowel disease to demonstrate that their results are consistent, regardless of the year o moment the patients were treated.

Author Response

Reviewer 3:

This study on the predictive capacity of 3 routinely used and widely available analytical markers in clinical practice is relevant, novel and employs appropriate methods. However, it starts from a premise that may not be completely true. The authors collect data from patients with inflammatory bowel disease treated with infliximab in the period 2008 to 2018. The criterion for failure to respond to treatment with infliximab consisted of a shift to another biological drug than infliximab or ceasing biological treatment altogether in the follow-up period of 32 weeks. Any patients who continued on infliximab beyond 32 weeks along the 10-year study period were considered to have responded to treatment, which is not necessarily true.

Answer: We agree and IBD is a capricious disease which can exacerbate at any time. This was one reason for choosing the physician’s assessment at the end of the induction period.  The patients who continue long-term treatments are not necessarily fully responding to the treatment, as the reviewer points out. It can be due to spontaneous remission, nevertheless it is based on clinical data and most patients will only be treated if the physician finds it clinically relevant.     

Adalimumab, the second biologic drug approved to treat Crohn's disease after infliximab, was approved in Europe for use in Crohn's disease in April 2007, but for ulcerative colitis, adalimumab was not approved by the European Union in April 2012. The next drug (vedolizumab) was not approved by the European Medicines Agency for both Crohn's disease and ulcerative colitis until May 2014. Finally, ustekinumab was approved in Europe to treat Crohn's disease in May 2016, and for ulcerative colitis in September 2019. Before these dates, biological alternatives to infliximab were simply not available for use in Europe.

Therefore, it seems logical to think that some patients included in the registry supporting this study could have maintained treatment with infliximab beyond 32 weeks, even if the effectiveness had been suboptimal, simply because there was no therapeutic alternative available at the moment they were treated. Taking this limitation into account, it would be necessary for the authors to develop subgroup analyses, considering various time frames and types of inflammatory bowel disease to demonstrate that their results are consistent, regardless of the year o moment the patients were treated.

Answer: Thank you for this and we agree that further sub-analyses are needed. In a sub-analysis in which the outcome was change to another biological drug only, the results were materially the same:

“In the analyses where the outcome was restricted to change to another biological drug than infliximab, all results were essentially the same as for the results in which stopping biological treatment was included in the outcome definition (data not shown).”

We have further conducted sub-analyses according to the reviewer’s suggestions. As only 5 patients were treated with ustekinomab in the follow-up period, we have ignored the dates for this drug. For adalimumab, we divided the follow-up period according to its start before or from 1 April 2012 and only analyzed patients with ulcerative colitis because adalimumab was approved for Crohn’s disease during the whole study period. For vedolizumab, we divided the follow-up period into before and from 1 May 2014, encompassing all patients with IBD. The results were materially the same as shown in Tables 2 and 3 (data not shown). We have added these sub-analyses to the article.      

Round 2

Reviewer 3 Report

Comments and Suggestions for Authors

The authors have tried to respond to criticism of the previous version of the work, without appreciating relevant changes in the content of the manuscript. In any case, the revised text has improved compared to the previous version.